# Properties and Microstructure Evaluation in NiAl-xWC (x = 0 − 90 wt.%) Intermetallic-Based Composites Prepared by Mechanical Alloying

**DOI:** 10.3390/ma16052048

**Published:** 2023-03-01

**Authors:** Daria Piechowiak, Albert Kania, Natalia Łukaszkiewicz, Andrzej Miklaszewski

**Affiliations:** Institute of Materials Science and Engineering, Poznań University of Technology, Jana Pawla II 24, 61-138 Poznan, Poland

**Keywords:** intermetallic-based composites, mechanical alloying, structural analysis, microstructure evaluation

## Abstract

In this work, NiAl-xWC (x = 0 − 90 wt.% WC) intermetallic-based composites were successfully synthesized by mechanical alloying (MA) and a hot-pressing approach. As initial powders, a mixture of nickel, aluminum and tungsten carbide was used. The phase changes in analyzed systems after mechanical alloying and hot pressing were evaluated by an X-ray diffraction method. Scanning electron microscopy and hardness test examination were used for evaluating microstructure and properties for all fabricated systems from the initial powder to the final sinter stage. The basic sinter properties were evaluated to estimate their relative densities. Synthesized and fabricated NiAl-xWC composites showed an interesting relationship between the structure of the constituting phases, analyzed by planimetric and structural methods and sintering temperature. The analyzed relationship proves that the structural order reconstructed by sintering strongly depends on the initial formulation and its decomposition after MA processing. The results confirm that it is possible to obtain an intermetallic NiAl phase after 10 h of MA. For processed powder mixtures, the results showed that increased WC content intensifies fragmentation and structural disintegration. The final structure of the sinters fabricated in lower (800 °C) and higher temperature regimes (1100 °C), consisted of recrystallized NiAl and WC phases. The macro hardness of sinters obtained at 1100 °C increased from 409 HV (NiAl) to 1800 HV (NiAl + 90% WC). Obtained results reveal a new applicable perspective in the field of intermetallic-based composites and remain highly anticipated for possible application in severe-wear or high-temperature conditions.

## 1. Introduction

Nickel aluminide (NiAl) is an intermetallic phase with potential applications in high-temperature structural materials and coatings due to its attractive properties [1]. Over a wide range of Ni content (45.0 to 58.5 at. % Ni), NiAl crystallizes in the A2 lattice (bcc) and exhibits an ordered L2o (B2) crystal structure (2 atoms per cell—1 nickel atom from the corners, 1 aluminum atom from the geometric center of the cell). Moreover, the melting point reaches 1911 K [2,3]. This ordered structure provides high-temperature strength and thermal stability, while the presence of aluminum in the compound significantly improves its oxidation resistance [4]. In addition, NiAl has high thermal conductivity, relatively low density (5.90 g/cm^3^) and relatively good tribological properties [5]. However, properties that often preclude the use of pure NiAl phase are its low ductility and high brittleness at room temperature, as well as low creep resistance at high temperatures [4,6].

To improve the properties of NiAl phase the composite approach could be used. The addition of other phases to a NiAl matrix (ceramics in particular) has been previously researched as it holds great potential for improving the mechanical properties of NiAl, such as wear resistance and hardness [7]. Various refractory and hard phases have been used as reinforcement, such as borides (CrB_2_-ZrB_2_ [8], ZrB_2_ [9], TiB_2_ [10] and TaB_2_-TaB [11]), nitrides (TiN [12]), oxides (Al_2_O_3_ [13,14], V_2_O_5_ [15] and Ag_3_VO_4_ [16]), and carbides (TiC-Al_2_O_3_ [17], Mo_2_C [18], NbC [19] and TiC [20], WC [5,21]). It has been confirmed that tribological properties could be improved significantly by the addition of the oxide as V_2_O_5_ nanowires, which allows a V_2_O_5_-enriched glaze film to form at elevated temperatures [15]. Another attempt was made with Ag_3_VO_4_ nanoparticles. In that case, the temperature-adaptive action of the phase composition led to the overall decrease in wear rate at a broad temperature range [16]. Moreover, the influence of the addition of graphene on the tribological properties of NiAl was also studied, where the formation of an anti-friction tribo-film has been observed on worn surfaces after the incorporation of graphene [22]. The addition of borides such as CrB_2_ and ZrB_2_ was recognized as an effective way to decrease the wear rate compared to NiAl with no reinforcement [8]. ZrB_2_ and TiB_2_ have been proven to significantly increase the microhardness of NiAl intermetallic phase [23]. Furthermore, an interesting example was recognized for WC, probably due to its well-known characteristics in cermets and sintered carbide applications. Various research has also been conducted on the addition of WC [4,5,6,21,24,25,26]. It is characterized by elevated hardness and wear resistance, so it can significantly strengthen NiAl composites. In addition, it reduces brittleness at low temperatures and increases creep resistance at high temperatures.

Previous studies of NiAl-WC systems have focused on the effects of WC on the microstructure and mechanical properties and obtained [5]. An example is the work on the effect of mechanical activation performed prior to combustion synthesis on the final microstructure of the composite. This study confirmed the relationship based on previous work indicating an increase in reactions’ kinetics during combustion synthesis with the application of mechanical activation [4]. Another work focused on alloyed coatings applied by laser surface alloying, which utilizes high-energy lasers to irradiate and melt precursor materials with the substrate surface. It studied the effects of WC and CeO_2_ additives on the tribological properties and behavior of composite coatings at temperatures of 25 °C and 400 °C. It also investigated the microstructure, microhardness and fracture toughness of the composite coatings [6] cladded on martensitic stainless steel. Research has proven that such prepared coatings (with a pre-and post-heat treatment) permit obtaining coatings that are free from cracks and pores. Due to the work-hardening characteristics of NiAl and WC’s strength, an increase in cavitation erosion resistance has been observed in the NiAl coating [24]. Thermal explosion reaction also remains interesting as a method to fabricate Ni intermetallic composites with WC. Starting with Ni, Al and WC powders, the reaction led to the fabrication of NiAl/WC composites with very good tribological properties in a short processing time [5,21]. Nanomaterials based on NiAl can also be successfully prepared by mechanical alloying with subsequent heat treatment [25]. 

The mechanical alloying (MA) method permits fabricating alloys and composites with a cold synthesis, for which the grain size could significantly diminish the number of generated defects [25]. It enables the synthesis of nanocrystalline materials, which may exhibit lower brittleness compared to their conventionally microcrystalline analogues [27]. 

The present study investigates NiAl-xWC composites (x = 0 wt.%, 10 wt.%, 20 wt.%, 40 wt.%, 60 wt.%, 80 wt.% and 90 wt.%) prepared by mechanical alloying of Ni, Al and WC powders, followed by hot-pressing sintering at 800 °C and 1100 °C. The conducted research takes into account the influence of the chemical composition, structural state of the prepared precursor powders and sintering temperature on the structure evaluation process and phase transformations during synthesis and the next sintering process.

## 2. Materials and Methods

### 2.1. Sample Preparation

The starting powder mixture was made from commercial powders of Ni (<1 μm, 99.5%, Sigma Aldrich, St. Louis, MO, USA), Al (99.8%, Onyxmet, Olsztyn, Poland) and WC (2 μm, 99%, Sigma Aldrich). Figure 1 shows SEM microphotographs of the starting components. The powders were mechanically alloyed to obtain intermetallic and composite precursors. The weight ratio of Ni and Al powders was 7:3 and the addition of WC was 10 wt.%, 20 wt.%, 40 wt.%, 60 wt.%, 80 wt.% and 90 wt.%. The powder precursors for all compositions were synthesized for 10 h in stainless steel vials under an argon atmosphere. The ball-to-powder ratio (BPR) equaled 10. The process was carried out on a SPEX 800 Mixer Mill (SPEX SamplePrep, Metuchen, NJ, USA). All activities related to the preparation of powders were conducted in a glove box with an inert argon atmosphere (Labmaster 130). 

In the next step, the powder’s precursors were consolidated through a hot pressing process. The sintering step was carried out on Elbit (Koszyce Wielkie, Poland) equipment at 800 °C and 1100 °C, with an acting pressure of 3 kN in a vacuum condition (<50 Pa). Heating was conducted through induction; the heating time was 30 s and the holding time at the sintering temperature was 300 s. The scheme of the sintering process is shown in Figure 2. The specimens prepared by the above procedure were marked as Table 1 shows.

### 2.2. Material Characterizations

The phase characterization of the powders during the mechanical alloying process and samples after consolidation was conducted using X-ray diffraction (XRD, Panalytical Empyrean, Almelo, Netherlands) equipment with a copper anode (CuKα—1.54 Å) with a Bragg–Brentano reflection mode. The measurement parameters were set up for voltage 45 kV, anode current 40 mA, 2θ range 20–90°, time per step 59.69 s/step and step size 0.0501° in all cases.

The following structural models were used for phase characterization:
Mechanically alloyed powders:
NiAl—ref. code 01-083-3994WC—ref. code 00-025-1047W_2_C—ref. code 01-079-5801
Sintered specimens:
NiAl—ref. code 04-005-7098WC—ref. code 04-016-4756



The determination of crystallite size and lattice strain of the phases after the mechanical alloying process was calculated by the Williamson–Hall method with an assumed uniform deformation model (UDM) used according to the formula below:$$\beta \cos\;\theta =\varepsilon \left(4\sin\;\theta \right)+\frac{K\lambda }{D}$$
where *β* is the width of X-ray diffraction at half maximum intensity, *θ* is the Bragg diffraction angle, *λ* is the wavelength of the radiation, *ε* is an inner strain, *D* is the crystallite size and *K* is the Scherrer constant. 

The mean internal strain can be obtained from the slope of *β*cos *θ* as a function of sin *θ* and the average crystallite size can be calculated from the intersection of this line at sin *θ* = 0. 

Lattice parameter estimation and phase quantitative analysis were based on the Rietveld profile fitting method performed using High Score Plus software with the PDF-4 ICDD structural database. The approach applied involved a simulation of the diffraction pattern based on the analyzed structural models listed above with reference codes. The calculated pattern of the model structure was fitted to the observed one by minimizing the sum of the squares and after a refinement using the Marquardt least squares algorithm. The definition of the residual pattern of the modelled data is as follows: Rwp—weighted pattern residual indicatorRexp—expected residual indicatorGOF—the goodness of fit


The powders’ morphology and sinters microstructure were characterized under different magnifications using a scanning electron microscope (SEM, MIRA3, Tescan, Brno, Czech Republic) supplied with an Ultimax 65 energy dispersive spectrometer detector (Oxford Instruments, Abingdon, Oxfordshire, UK). To compare the results obtained by the Rietveld method, the percentage of the NiAl phase in the composites sintered at 1100 °C was determined by the planimetric method. Olympus Stream phase analysis software was used for this purpose. The particle-size analysis of the powders was performed using the MountainsSEM software (Digital Surf, Besançon, France) based on the SEM microphotographs.

Vickers hardness (HV) was measured based on the ISO 6507-1 standard, using a microhardness tester (INNOVOTEST Nexus 4302, Maastricht, The Netherlands). For each polished sample, 10 measurements were made along the cross-section of the sample under a load of 1 kg. The load operating time was 10 s. 

The density of the sinters was determined using the Archimedes method in deionized water at 25 °C, while the theoretical density of the materials was calculated for the rule of the mixtures.

## 3. Results and Discussion

In the present study, the composites based on an intermetallic NiAl matrix with WC reinforcement were obtained using mechanical alloying (MA) and powder metallurgy (PM) methods. 

In the next step, the impact of the ceramic phase addition on phase transformation, microstructure and such properties as hardness and density were determined.

### 3.1. Structural and Morphological Powder Analysis

#### 3.1.1. Synthesis of the Intermetallic NiAl matrix

The process of mechanical alloying allowed the synthesis of the NiAl intermetallic phase. Figure 3 summarizes the XRD results for the Ni-Al system after different synthesis times. It can be seen that the XRD pattern after 15 min of milling shows sharp peaks, which indicate the raw Al and Ni phases. After 2 h of the process, the transformation to the NiAl intermetallic phase begins; however, unreacted starting components remain in the composition. With increasing milling time up to 5 h, peaks originating from the Al and Ni phase recede. Further milling causes a decrease in the intensity and an increase in half-widths of the peaks, which proves the fragmentation of crystallites and ongoing amorphization. In work [1], the NiAl phase was also fabricated by mechanical synthesis; however, in this case the synthesis of the phase took place after 30 h of grinding. However, Zarezadeh Mehrizi et al. [2] reported the formation of the NiAl phase after 10 h of milling. In this work, it was possible to obtain the NiAl phase after 5 h of grinding, which increases the efficiency of the synthesis process.

The size of crystallites and stresses at various stages of the synthesis were determined according to the Willamson –Hall equation. The results are presented in Table 2 and Figure 4. 

After 1 h of milling, the crystallite size was 72 nm and was reduced to 10 nm after 10 h of synthesis. 

Figure 1 shows SEM microphotographs of the Al, Ni and WC powders. The starting components differ significantly in both morphology and particle size. Al powder is characterized by a flake morphology, the average size of which was 21.85 µm. However, in the case of Ni and WC the powders are characterized by an irregular morphology; their sizes were 0.64 µm and 3.17 µm, respectively.

The SEM microphotograph of the NiAl powder after 10 h MA is shown in Figure 5. This powder was characterized by a large irregularity in the shape and dimensions of the particles, the average size of which was 2.05 µm; the standard deviation is 0.83 µm.

#### 3.1.2. Synthesis of NiAl-WC Composite Systems

Figure 6 shows the XRD patterns of the powders after 10 h of mechanical alloying for various WC contents. After the process, the composite systems contain NiAl, WC phases and a small amount of W_2_C. However, the W_2_C phase occurs only in the powder composition. The phase recedes after the sintering process. The formation of the W_2_C phase after the mechanical alloying process means that during grinding the WC is depleted of carbon, which diffuses into the defective NiAl phase. During sintering, recrystallization of the phases and re-diffusion of carbon into the WC phase take place. 

Willamson–Hall analysis was also carried out for the composite powders. The obtained results are presented in Figure 7 and Table 3. The size of the crystallites was determined for the peaks originating from the WC phase. Due to the coverage of most of the NiAl phase peaks by the WC phase peaks, some of the peaks were not included in the analysis. The size of the crystallites ranged from 21 mm to 38 mm after 10 h of milling. It can be noticed that the intensity of fragmentation of the NiAl phase increases with the increase in the content of the reinforcement phase. Analyzing the 44.638° peak of the NiAl phase with the highest intensity, it can be seen that its half-width increases from 1.375° for 0% WC to 1.85° for 60% WC; for higher WC contents the parameter cannot be measured.

SEM microphotographs of the NiAl-WC composite powders are shown in Figure 8. All compositions are characterized by irregular particle shapes. Moreover, it can be observed that the tendency of the powders to agglomerate increased as the WC content increased. The average powder particle size was measured from SEM micrographs and is presented in Table 4. The greatest fragmentation of the powder particles was obtained for the NiAl-10WC composition, while the NiAl-80WC powder was characterized by the greatest tendency to agglomerate.

### 3.2. Structural and Microstructural Analysis of Bulk Composites

Phase analysis after the sintering process at 800 °C and 1100 °C is shown in Figure 9. After the hot-pressing process, regardless of the temperature, the composites consisted of NiAl and WC phases. A comparable phase composition was achieved in work [2] after thermal stabilization of the NiAl-WC system, where the powder composition of NiAl-WC after 40 h of milling was annealed at 1100 °C for 1 h. 

XRD analysis revealed that the lower sintering temperature results in limited diffusion of components, which reduces grain growth. In particular, the discrepancy between temperatures starts to be apparent for compositions from 40% WC content as a broadening of peaks. This relationship is also observed in SEM microphotographs (Figure 10). 

SEM analysis revealed the influence of composition and temperature on the system’s ability to recrystallize and consolidate. The first differences in the morphological structure of the system are noticeable for 60% WC, while for 80% WC they are significant. Changes in microstructure mainly concern the grain morphology of the WC phase and its size range. Differences can be found, firstly, between the temperature of sintering, and secondly, between individual compositions. 

For the higher sintering temperature (1100 °C), an increase in the content of the WC additive between 40% and 60% transforms the morphological character of the reinforcement phase. For 60% of WC, the morphology of the grains begins to have a polygonal shape; the trend continues with the increase in WC content. In addition, for 80% and 90%, there is a strong grain growth of WC reinforcement phase. The structure of WC grains in systems with a high additive content coagulates, decreasing in the same dispersion of the reinforcing phase.

The microstructure of the obtained composites is also affected by the lower (800 °C) consolidation temperature for systems above 60% WC. Visible porosity at the boundaries of the consolidated powder particles as well as intermetallic matrix phase separation could be distinguished. For applied higher temperature regimes, porosity appears at the phase grain boundary region, which suggests its movement. Composites sintered at 800 °C are characterized by a much finer grain compared to those sintered at a higher temperature; the dispersion reinforcement mechanism stays dominant. This proves that the energy of the system for recrystallization could be too low for diffusion, which is blocked. 

An increase in sintering temperature allows the diffusion of components and a high level of structure strain leads to observed grain growth. However, this relation only occurs with a high content of the WC additive. The increase in temperature at contents below 40% does not affect the change in morphology and the growth of WC phase, indirectly pointing to the relation between the structure strain level and the analyzed composition.

The relationship between the composition and the tendency to evaluate the structure can be found in the stress state of powder systems after mechanical synthesis. As the WC content increases, the stresses in the structure increase (see Table 4) from 0.0051 for 10% to 0.77 for 90%. The increase in stress translates into lower energy needed to initiate active recrystallization and grain growth; however, for the lower sintering temperature it may be blocked by the growing reinforcement phase amount.

Following the structural calculation, the estimated phase amounts (PA) collated in Table 5 show a close relation to the planimetric analysis and initial compositions. The results, confirmed with a goodness of fit parameter (GOF), show lower reinforcement phase amounts for lower sintering temperatures until the relationship reaches 40 wt.% of WC. Higher amounts of reinforcing phase in precursor powders result in its growing relationship with higher sintering temperature for the obtained sinters.

### 3.3. Hardness and Density of NiAl-WC Composites

Hardness test results are presented in Table 6 and Figure 11. In the case of samples sintered at 800 °C, the hardness increased up to 80% WC content, followed by a decrease for 90% WC. The decrease in hardness is caused by the increase in porosity for higher contents of the reinforcing phase. In the case of samples sintered at 1100 °C, the hardness increased up to 90% WC. Moreover, it can be observed that samples sintered at 800 °C are characterized by a higher hardness up to 60% WC. This is due to the strengthening of grain boundaries. Above 60% WC, the hardness is more favorable for sinters at 1100 °C, where porosity starts to affect the result.

Comparing the calculated densities with the theoretical ones in Table 6, it can be seen that the calculated densities are in most cases higher than the theoretical ones. This is due to the limitations of the analytical method based on the rule of the mixtures, which takes into account the content of individual components and their density. However, it is used to indicate the trend of density changes depending on the content of the reinforcing phase.

Density values for samples sintered at 800 °C and 1100 °C are comparable up to 40% of WC phase. A further increase in the carbide content causes a decrease in density below the theoretical one for systems sintered at 800 °C. At the sintering temperature of 1100 °C, the density values are close to the theoretical ones (see Figure 12 and Table 6). 

For systems with a WC phase content of up to 40%, the best set of properties was obtained with a sintering temperature of 800 °C. The sinters are characterized by fine grain in the submicron range, high hardness and high density. The increase in the content of WC phase above 40% forces the use of higher sintering temperatures to obtain sinters with higher density, which translates into an increase in their hardness.

## 4. Conclusions

In this work, composites based on NiAl intermetallic phase with various WC contents were synthesized using a mechanical alloying process and a hot-pressing approach. Based on the conducted research, the following conclusions can be drawn:
-The mechanical alloying process allows the synthesis of intermetallic phase from pure Al and Ni elements. After 5 h of milling, a new phase appeared in composition;-The addition of WC phase did not affect the formation of NiAl intermetallic phase;-The addition of WC increased the intensity of the milling process respective to the analyzed structure strain level;-After the hot-pressing process, the phase composition of composites consists of NiAl and WC phases;-The reinforcing phase amount and final structure properties strongly depend on the initial composition, its structural state and applied sintering conditions influencing the dominative reinforcement mechanism; -The microstructure of the obtained composites shows an ultrafine-grain range;-The composites are characterized by elevated hardness, which reached 1800 HV for sample 90–1100.


## Figures and Tables

**Figure 1 materials-16-02048-f001:**
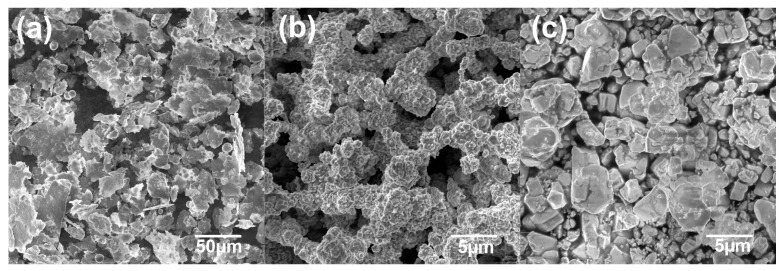
SEM microphotographs of Al (**a**), Ni (**b**) and WC (**c**) powder precursors.

**Figure 2 materials-16-02048-f002:**
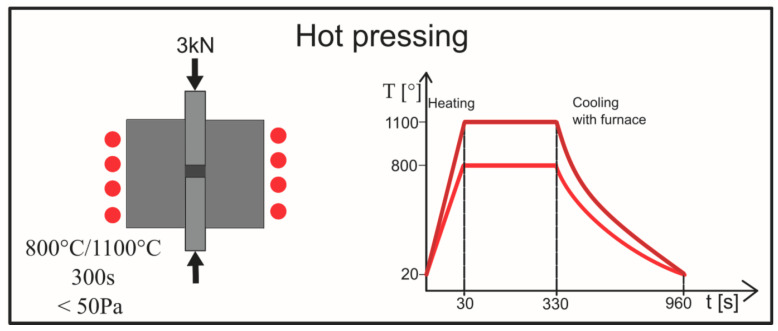
Schematic illustration of the hot-pressing process.

**Figure 3 materials-16-02048-f003:**
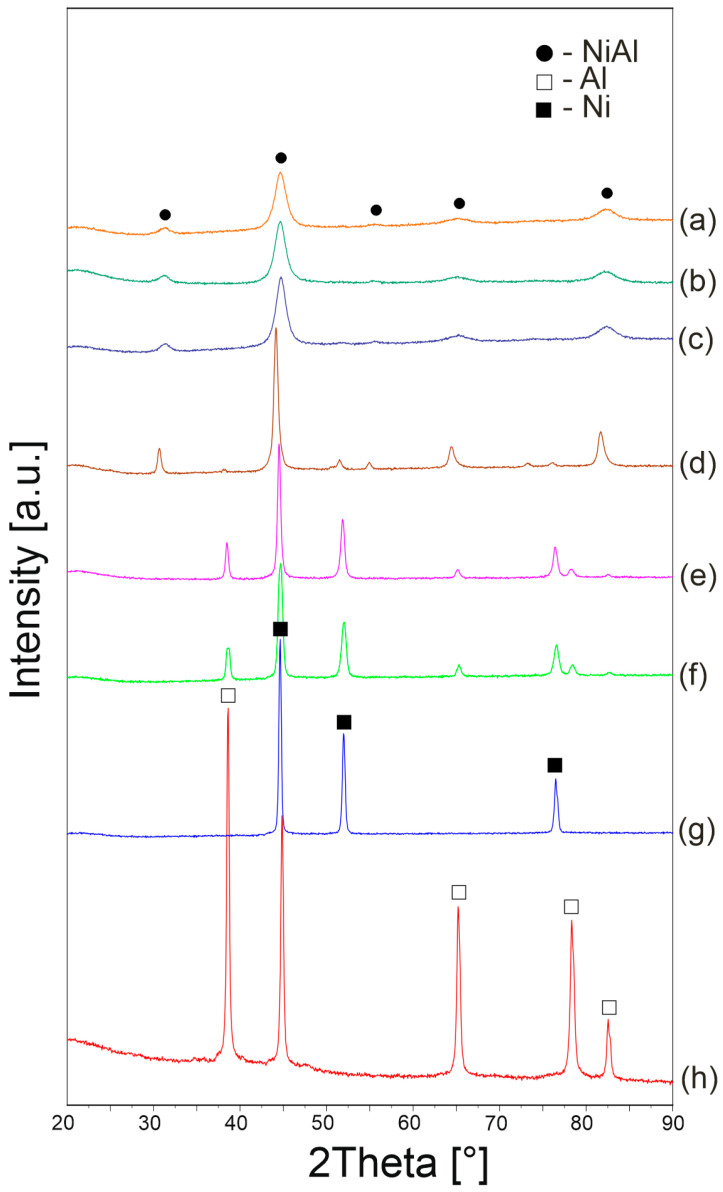
XRD patterns of NiAl powder after 10 h (a), 7 h (b), 5 h (c), 2 h (d), 1 h (e) and 15 min (f) of MA and XRD patterns of pure Ni (g) and Al (h).

**Figure 4 materials-16-02048-f004:**
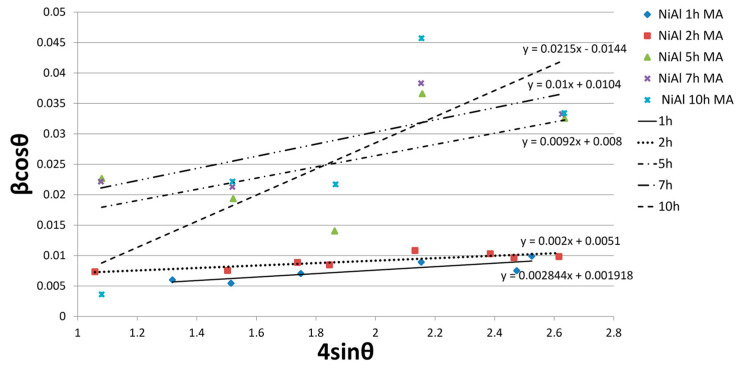
Linear Williamson–Hall plots with an obtained equation based on the XRD spectra of NiAl powder after various milling times.

**Figure 5 materials-16-02048-f005:**
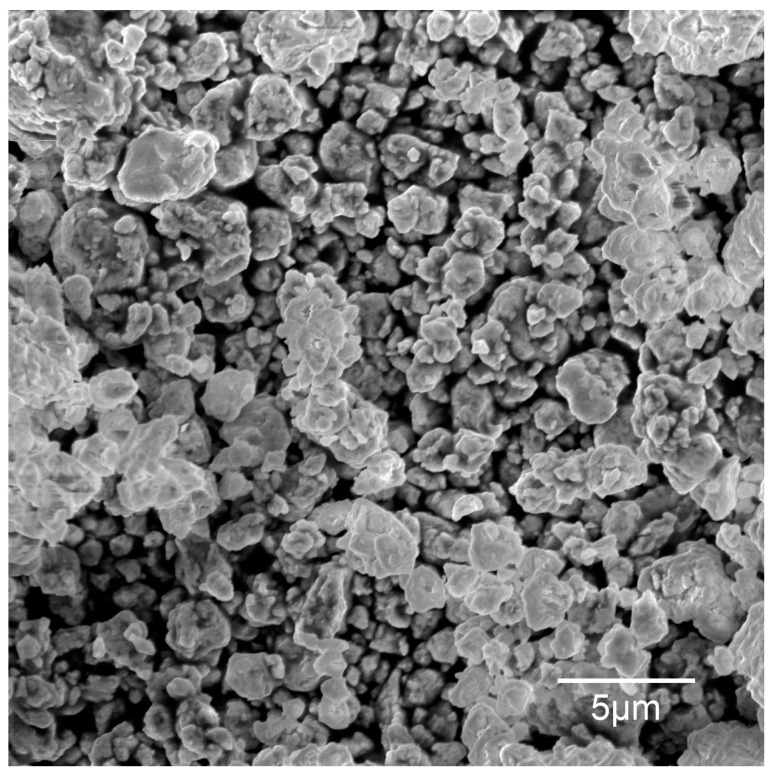
SEM microphotographs of NiAl powder after 10 h of MA.

**Figure 6 materials-16-02048-f006:**
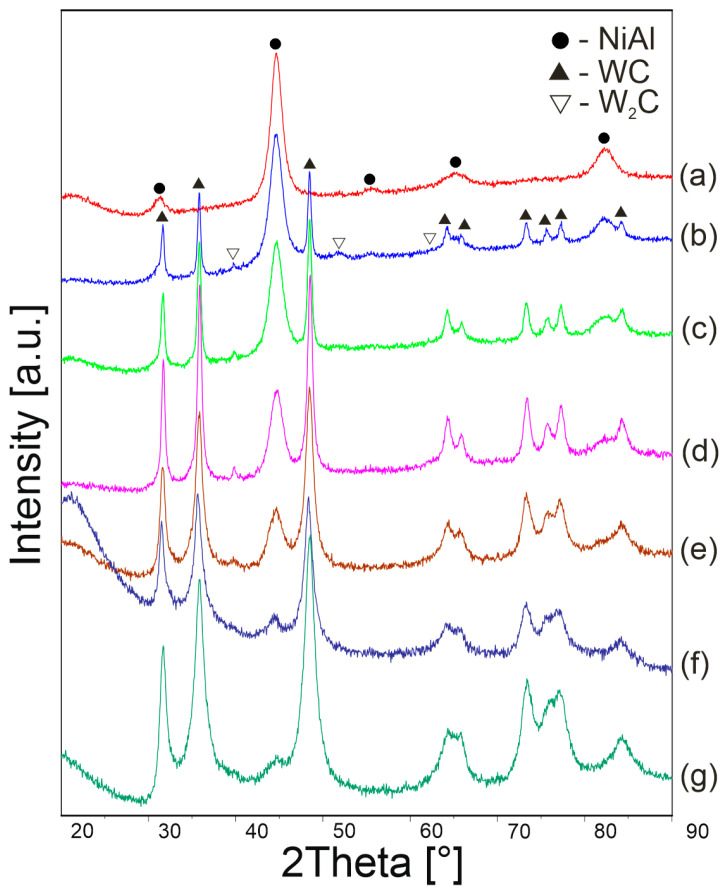
XRD patterns of NiAl (a), NiAl-10WC (b), NiAl-20WC (c), NiAl-40WC (d), NiAl-60WC (e), NiAl-80WC (f) and NiAl-90WC (g) powders after 10 h of MA.

**Figure 7 materials-16-02048-f007:**
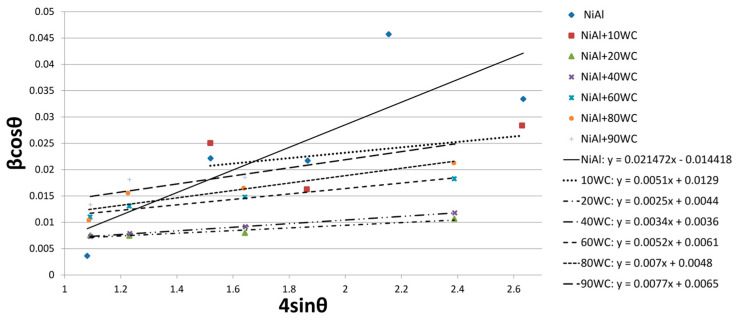
Linear Williamson–Hall plots with an obtained equation based on the XRD spectra of NiAl-xWC powder after 10 h of mechanical alloying.

**Figure 8 materials-16-02048-f008:**
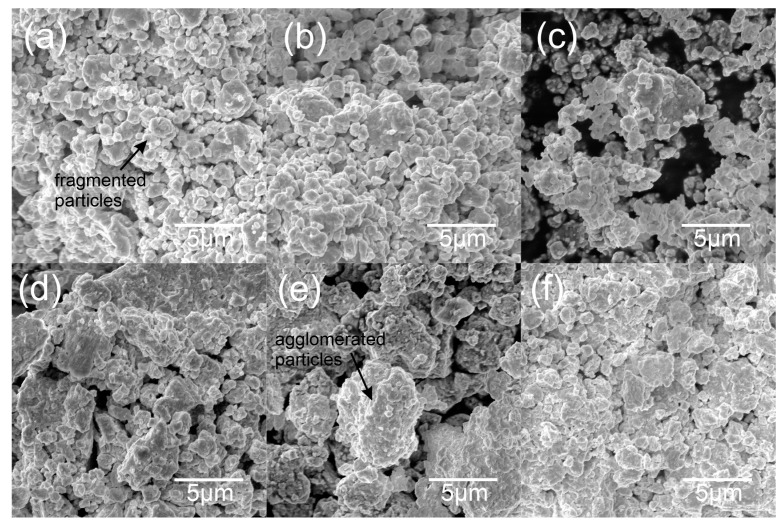
SEM microphotograph of NiAl-10WC (**a**), NiAl-20WC (**b**), NiAl-40WC (**c**), NiAl-60WC (**d**), NiAl-80WC (**e**) and NiAl-90WC (**f**) powders after 10 h of MA.

**Figure 9 materials-16-02048-f009:**
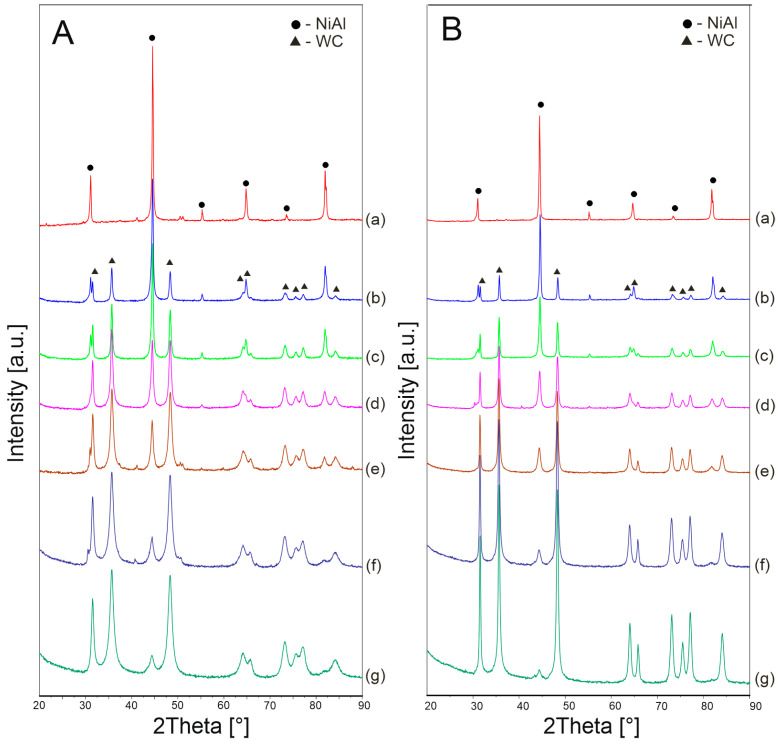
(**A**): XRD patterns of NiAl-800 (a), 10–800 (b), 20–800 (c), 40–800 (d), 60–800 (e), 80–800 (f) and 90–800 (g). (**B**): XRD patterns of NiAl-1100 (a), 10–1100 (b), 20–1100 (c), 40–1100 (d), 60–1100 (e), 80–1100 (f) and 90–1100 (g).

**Figure 10 materials-16-02048-f010:**
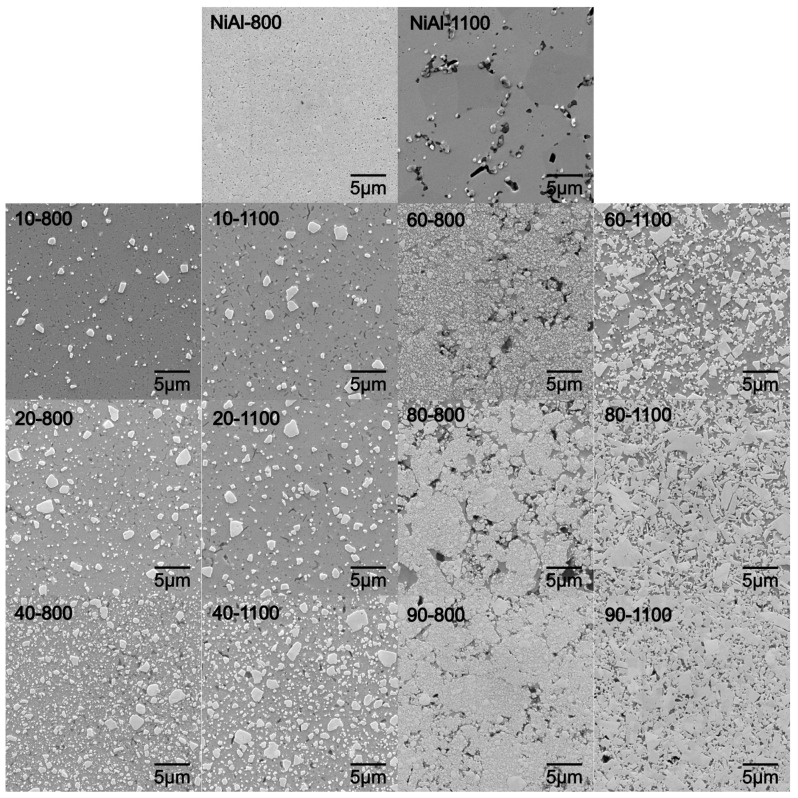
SEM microphotographs of NiAl-xWC samples sintered at 800 °C and 1100 °C.

**Figure 11 materials-16-02048-f011:**
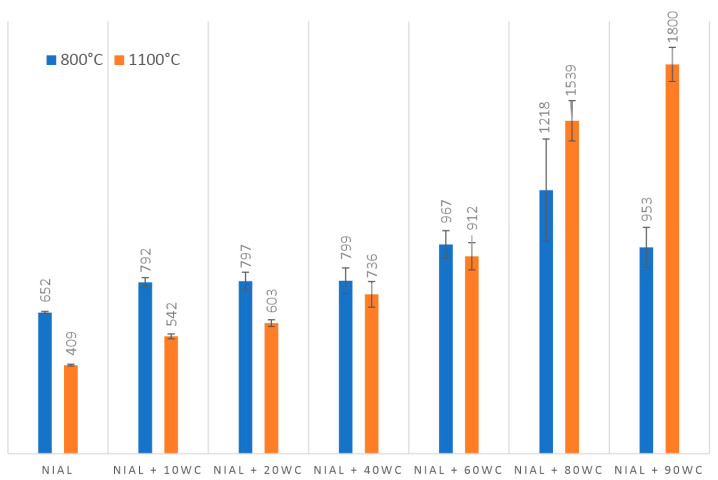
Vickers hardness of NiAl-xWC composites sintered at 800 °C and 1100 °C.

**Figure 12 materials-16-02048-f012:**
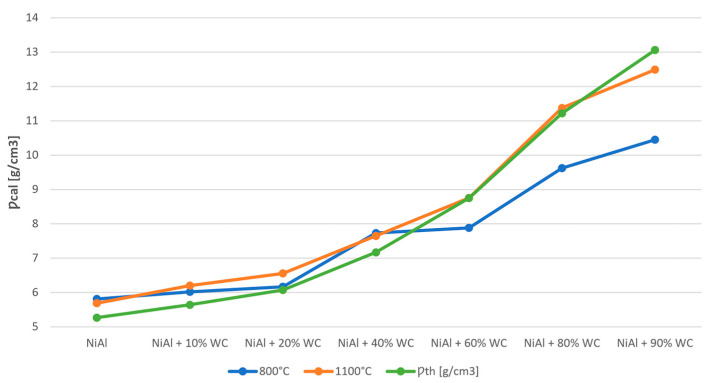
Calculated density and theoretical density of NiAl-xWC composites sintered at 800 °C and 1100 °C.

**Table 1 materials-16-02048-t001:** Bulk sample indications.

Sample	Sintering Temperature [°C]	Symbol
NiAl	800	NiAl-800
NiAl + 10% WC		10–800
NiAl + 20% WC		20–800
NiAl + 40% WC		40–800
NiAl + 60% WC		60–800
NiAl + 80% WC		80–800
NiAl + 90% WC		90–800
NiAl	1100	NiAl-1100
NiAl + 10% WC		10–1100
NiAl + 20% WC		20–1100
NiAl + 40% WC		40–1100
NiAl + 60% WC		60–1100
NiAl + 80% WC		80–1100
NiAl + 90% WC		90–1100

**Table 2 materials-16-02048-t002:** Estimated structure size and strain factors based on Williamson–Hall plots for NiAl powders after various milling times.

NiAl Milling Time	ε	D [nm]
1 h	0.002844	72
2 h	0.002012	27
5 h	0.009218	17
7 h	0.009970	13
10 h	0.021500	10

**Table 3 materials-16-02048-t003:** Estimated size and strain factors based on Williamson–Hall plots for NiAl-xWC powders after 10 h of mechanical alloying.

Powder Composition	ε	D [nm]
NiAl + 10% WC	0.0051	28
NiAl + 20% WC	0.0025	32
NiAl + 40% WC	0.0034	39
NiAl + 60% WC	0.0052	23
NiAl + 80% WC	0.0070	29
NiAl + 90% WC	0.0077	21

**Table 4 materials-16-02048-t004:** Average particle size of powders measured based on SEM microphotographs.

Powder Composition	Average Powder Particle Size [µm]	Standard Deviation [µm]
Al	21.85	13.16
Ni	0.64	0.34
WC	3.17	1.13
NiAl	2.05	0.83
NiAl + 10% WC	1.13	0.68
NiAl + 20% WC	1.45	0.63
NiAl + 40% WC	1.76	0.99
NiAl + 60% WC	2.29	0.96
NiAl + 80% WC	3.10	1.83
NiAl + 90% WC	1.75	0.63

**Table 5 materials-16-02048-t005:** Structural phase parameters and amounts determined by the Rietveld method collated with the planimetric phase estimation.

Sample	NiAl			WC				Rwp [%]	Rexp [%]	GOF	Planimetric NiAl Content [%]
	a [Å]	V [Å]	PA [%]	a [Å]	c [Å]	V [Å]	PA [%]				
NiAl-800	2.877748	23.83189	100	-	-	-	-	2.79074	1.26216	3.45235	-
NiAl-1100	2.881795	23.93257	100	-	-	-	-	3.52871	1.53387	4.15852	-
10–800	2.877455	23.82459	91.4	2.905589	2.837235	20.74408	8.6	3.88914	1.64960	3.72663	-
10–1100	2.876814	23.80868	94.0	2.904611	2.836464	20.72449	6.0	3.20058	1.66355	2.99967	90.2
20–800	2.878794	23.85789	82.0	2.905391	2.835926	20.73169	18.0	3.04924	1.72062	2.80676	-
20–1100	2.879063	23.86455	82.5	2.906711	2.836687	20.75611	17.5	3.80800	1.69231	3.69920	82.0
40–800	2.883199	23.96756	59.7	2.907169	2.837852	20.77116	40.3	3.78441	1.89541	2.88086	-
40–1100	2.885629	24.02822	59.5	2.910031	2.839636	20.82517	40.5	3.87954	1.92495	3.08873	65.1
60–800	2.884665	24.00413	42.9	2.905388	2.836926	20.73895	57.1	4.58649	2.15801	2.76792	-
60–1100	2.889674	24.12940	35.9	2.911327	2.839604	20.84349	64.1	4.94694	2.24004	3.37547	53.3
80–800	**-**	-	-	-	-	-	-	-	-	-	-
80–1100	2.891775	24.18207	14.3	2.911662	2.839584	20.84814	85.7	6.816	2.69761	3.23206	22.5
90–800	**-**	-	-	-	-	-	-	-	-	-	-
90–1100	2.890064	24.13917	13.2	2.910384	2.839364	20.82823	86.8	7.65265	2.73152	3.37985	11.7
Ref. WC	-	-	-	2.9063	2.8375	20.76	-	-	-	-	-
Ref. NiAl	2.8855	24.02	-	-	-	-	-	-	-	-	-

**Table 6 materials-16-02048-t006:** Vickers hardness (HV), calculated density (Ƿcal) and theoretical density (Ƿth) of sinters.

		NiAl	NiAl + 10WC	NiAl + 20WC	NiAl + 40WC	NiAl + 60WC	NiAl + 80WC	NiAl + 90WC
800 °C	HV	652 ± 5	792 ± 21	797 ± 41	799 ± 59	967 ± 63	1218 ± 237	953 ± 93
	Ƿcal [g/cm^3^]	5.81	6.02	6.17	7.73	7.88	9.62	10.45
1100 °C	HV	409 ± 4	542 ± 11	603 ± 16	736 ± 59	912 ± 63	1539 ± 93	1800 ± 79
	Ƿcal [g/cm^3^]	5.69	6.20	6.56	7.65	8.75	11.37	12.49
Ƿth [g/cm^3^]		5.27	5.64	6.08	7.17	8.75	11.22	13.06

## Data Availability

Not applicable.

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
