# Peer review of "Properties and Microstructure Evaluation in NiAl-xWC (x = 0 − 90 wt.%) Intermetallic-Based Composites Prepared by Mechanical Alloying"

_materials, 2023, doi:10.3390/ma16052048_

Round 1
Reviewer 1 Report
This study does not apply clear novelty. Encourage highlighting details of the relationship between microstructure and properties e.g purpose mechanism.
Major to the manuscript also should be taken out such as
1. Manuscript should be written using a scientific approach
2. The quality of the figures e.g labeling, wrong axis, figure captions
3. The citations and references.
Author Response
Dear Reviewer 1,
We would like to thank you for your comments and kind decision for us to resubmit our manuscript: Properties and microstructure evaluation in NiAl-xWC (x=0-90wt.%) intermetallic-based composites prepared by mechanical alloying to Materials. We are pleased with your comments and objective feedback considering the draft of our article. The changes were marked in our article in red colour.
All of the suggestions were included during the correction of our article and the comments were appropriately responded:
This study does not apply clear novelty. Encourage highlighting details of the relationship between microstructure and properties e.g purpose mechanism.
Thank you for this comment. The highlights were added to the manuscript.
Major to the manuscript also should be taken out such as
- Manuscript should be written using a scientific approach
The authors agree with that comment, a proper improvement was done in the manuscript.
- The quality of the figures e.g labeling, wrong axis, figure captions
The authors agree with that comment, a proper improvement was done in the manuscript.
- The citations and references.
The authors agree with that comment, a proper improvement was done in the manuscript.
We would like to thank you once again for your suggestions for improving our manuscript.
Yours faithfully, Miklaszewski
Reviewer 2 Report
The subject and object of the presented research are actual, and correspond to the aims and topics of the journal. However, the presentation of the materials is very confusing and difficult to read. Thus, it is impossible to fully evaluate the results obtained. The manuscript cannot be published in its present form; it needs to be reorganized and thoroughly revised. Below are a few comments and suggestions.
P1. NiAl-xWC: explain what “-x” means.
P2. I strongly recommend revising the language and style of the manuscript, as it contains phrases and sentences that are difficult to read.
P3. Line 32. “NiAl exhibits an ordered L2o (B2)…” – it is necessary to explain shortly what “L2o” means as well as B2 for the benefit of the readers.
P4. Please give references in the text according to the "Instructions to Authors. Figure captions should follow the figures.
P5. Figure references in the text are unreadable (see, for example, lines 101, 114, etc.).
P6. The authors write that there are other works in which the NiAl phase was obtained using MA, but the procedure requires more time compared to the presented work. In my opinion, the observed cases need to be discussed briefly.
P7. It would be appreciated to include EDX data in the manuscript.
P8. Table 5: standard deviations are necessary.
P9. It appears that Figure 10 is actually "SEM micrographs of NiAl-xWC samples sintered at 800°C and 1100°C," and one of the figures is missing.
P10. It is necessary to highlight why further research on these subjects is potential, based on the brief literature review in the "Introduction" section. The authors should also describe their main results in comparison with the literature.
Author Response
Dear Reviewer 2,
We would like to thank you for your comments and kind decision for us to resubmit our manuscript: Properties and microstructure evaluation in NiAl-xWC (x=0-90wt.%) intermetallic-based composites prepared by mechanical alloying to Materials. We are pleased with your comments and objective feedback considering the draft of our article. The changes were marked in our article in red colour.
All of the suggestions were included during the correction of our article and the comments were appropriately responded:
P1. NiAl-xWC: explain what “-x” means.
We appreciate that comment. The “-x” means the weight percentage content of WC phase added to systems which are 0 wt.%, 10 wt.%, 20 wt.%, 40 wt.%, 60 wt.%, 80 wt.%, 90 wt.%. We clarified the designation in the text. (see abstract and p. 2 l. 86-87)
P2. I strongly recommend revising the language and style of the manuscript, as it contains phrases and sentences that are difficult to read.
Thank you for this comment the final text was revised by the native speaker for the improvements.
P3. Line 32. “NiAl exhibits an ordered L2o (B2)…” – it is necessary to explain shortly what “L2o” means as well as B2 for the benefit of the readers.
Thank you for this comment. We added proper information in the text. (see p1. L. 32-34)
P4. Please give references in the text according to the "Instructions to Authors. Figure captions should follow the figures.
The authors agree with that comment, a proper improvement was done in the manuscript
P5. Figure references in the text are unreadable (see, for example, lines 101, 114, etc.).
Thank you for your suggestion. We have corrected that mistake.
P6. The authors write that there are other works in which the NiAl phase was obtained using MA, but the procedure requires more time compared to the presented work. In my opinion, the observed cases need to be discussed briefly.
Thank you for your suggestion however, all necessary data for MA processing were included in the methodological subsection. The observed difference outcome is a difference in processing parameters.
P7. It would be appreciated to include EDX data in the manuscript.
Thank you for your suggestion. We planned to do an EDX analysis in the next publication focusing on the direct diffusion relation in the structure.
P8. Table 5: standard deviations are necessary.
Thank you for your suggestion. We added crystallographic parameters of reference phases used in the Rietveld analysis (see Table 5). The applied fitting method allows determining the Rwp Rexp and GOF parameters which represent the goodness of estimated values to collected data.
P9. It appears that Figure 10 is actually "SEM micrographs of NiAl-xWC samples sintered at 800°C and 1100°C," and one of the figures is missing.
We appreciate that comment. We corrected the figure captions and added the missing figure. (see Figure 12)
P10. It is necessary to highlight why further research on these subjects is potential, based on the brief literature review in the "Introduction" section. The authors should also describe their main results in comparison with the literature.
Thank you for this comment. The highlights were added to the manuscript
We would like to thank you once again for your suggestions for improving our manuscript.
Yours faithfully, Miklaszewski
Reviewer 3 Report
The manuscript is about the properties and microstructure evaluation in NiAl-xWC (x=0-90wt.%) intermetallic-based composites.
First of all, in the manuscript, there are many warning messages as “Error! Reference source not found”. Therefore, I will request to review the manuscript again after revision.
And please consider following comments:
(1) In introduction part, authors say “The present study investigates NiAl-based composites with different WC content …..”
Any alloy-based composites include max % 50 reinforcements. In present study, with WC content (60 wt.%, 80 wt.%, 90 wt.%), over %50 WC reinforcements, they are not NiAl based composite. They are WC based NiAl reinforced composite. Therefore, the results in (60 wt.%, 80 wt.%, 90 wt.%) WC content must be completely removed.
(2) It is strongly recommended authors to add visual research gap and identify the novelty of the manuscript. It can be placed after introduction section as a separate section.
(3) Image sizes are very big in many Figures. Examples, “Figures 3, 5, 6”. Titles and Figures in separate pages.
(4) Figure 2 , in cooling curve, temperature decreases from 1100 oC to room temperature (20 oC) in a short time (between 330 s and 960 s). Is cooling process applied in hot pressing system or in air condition? How did authors cool the mould in a short time?
(5) There are some standard methods to measure powder particle size. What kind of method was applied by authors? Please give detail for Table 3.
(6) Authors say “determination of crystallite size and lattice strain of the phases was calculated by the Williamson–Hall method … used according to the formula below”.
?????=?(4????)+?? /?
I suggest authors to check calculated Linear Williamson-Hall plots values in Figure 8.
Can authors give an example about calculating for any chosen value?
(7) In section 3.3, please check the first and second sentences. Sentence is incomplete.
“The hardness test results are presented in Table 6 and ”
(8) There are 14 images in Figure 10. If it is presented in 4 columns, it will be better. Examining will be easier.
Author Response
Dear Reviewer 3,
We would like to thank you for your comments and kind decision for us to resubmit our manuscript: Properties and microstructure evaluation in NiAl-xWC (x=0-90wt.%) intermetallic-based composites prepared by mechanical alloying to Materials. We are pleased with your comments and objective feedback considering the draft of our article. The changes were marked in our article in red colour.
All of the suggestions were included during the correction of our article and the comments were appropriately responded:
(0) First of all, in the manuscript, there are many warning messages as “Error! Reference source not found”. Therefore, I will request to review the manuscript again after revision.
The authors agree with that comment. We have corrected that mistake.
(1) In introduction part, authors say “The present study investigates NiAl-based composites with different WC content …..”
Any alloy-based composites include max % 50 reinforcements. In present study, with WC content (60 wt.%, 80 wt.%, 90 wt.%), over %50 WC reinforcements, they are not NiAl based composite. They are WC based NiAl reinforced composite. Therefore, the results in (60 wt.%, 80 wt.%, 90 wt.%) WC content must be completely removed.
Thank you for this comment. We understand your point of view regarding nomenclature and agree that over 50% of the WC phase is overwhelming and is the main phase. However, the terminology used for composite structures in many examples points out the matrix and reinforcement phase, suggesting when something is based on, then is a matrix - in our example an intermetallic phase. The conducted research has a strong scientific impact in the present form and presents a wide spectrum of influence of particular phases on the evaluation of structure and properties, which from a scientific point of view is much more important than introducing additional designations that may confuse the reader. Removing the results would drastically reduce the value of the article and we don’t feel that is necessary.
(2) It is strongly recommended authors to add visual research gap and identify the novelty of the manuscript. It can be placed after introduction section as a separate section.
Thank you for this comment. The authors decide to add a highlight to the manuscript
(3) Image sizes are very big in many Figures. Examples, “Figures 3, 5, 6”. Titles and Figures in separate pages.
We appreciate that comment. We corrected the figure and figure captions.
(4) Figure 2 , in cooling curve, temperature decreases from 1100 oC to room temperature (20 oC) in a short time (between 330 s and 960 s). Is cooling process applied in hot pressing system or in air condition? How did authors cool the mould in a short time?
Thank you for this comment. Cooling is carried out in the press system in which the induction coil is cooled in vacuum conditions. Thanks to this, it is possible to quickly dissipate heat through the system. Additionally, the whole processing chamber possesses an inside cooling flow which protects the chamber and allows for a decrease in the temperature quicker inside the working area. You may also see the schematic arrangement of the system in our other works. The problems with cooling rise with die dimensions and over 1500°C, then the flow of the coolant need to be increased to protect the coil.
https://doi.org/10.3390/ma12040653
https://doi.org/10.1016/j.ijrmhm.2016.10.007
https://doi.org/10.1016/j.jallcom.2018.10.217
https://doi.org/10.1016/j.jallcom.2018.06.224
(5) There are some standard methods to measure powder particle size. What kind of method was applied by authors? Please give detail for Table 3.
Thank you for this comment. The particle size analysis of the powders was performed using the MountainsSEM software (Digital Surf) based on the computing of the SEM microphotographs, the method is not standardized because its an invention of the software company (see p. 4 l 169-170).
(6) Authors say “determination of crystallite size and lattice strain of the phases was calculated by the Williamson–Hall method … used according to the formula below”.
?????=(4????)+?? /?
I suggest authors to check calculated Linear Williamson-Hall plots values in Figure 8.
Can authors give an example about calculating for any chosen value?
Thank you for this comment. We notice the mistakes. We calculated crystallite size and lattice strain in the following method:
W-H formula was compared to the trend line, where:
?????=(4????)+?? /?
y=ax+b
y=?????
x=(4????)
a= ?
b= ?? /?
D= ?? /b
|
βhkl [˚] |
2 ? |
cos(?) |
sin(?) |
βhkl*cos(?) |
4sin(?) |
e=A |
b=(Kl)/D |
D=(Kl)/b |
|
0,216 |
31,347 |
0,9628168 |
0,2701552 |
0,003629734 |
1,080621 |
0,025 |
0,0144 |
9,625 |
|
1,372 |
44,665 |
0,9249943 |
0,3799809 |
0,022149838 |
1,519924 |
|||
|
1,406 |
55,614 |
0,884524 |
0,4664947 |
0,021705625 |
1,865979 |
|||
|
3,109 |
65,177 |
0,8425605 |
0,5386017 |
0,04571926 |
2,154407 |
|||
|
2,544 |
82,361 |
0,752639 |
0,6584333 |
0,033418059 |
2,633733 |
(7) In section 3.3, please check the first and second sentences. Sentence is incomplete.
“The hardness test results are presented in Table 6 and ”
We appreciate that comment, the text was changed. (see. P12 l. 350)
(8) There are 14 images in Figure 10. If it is presented in 4 columns, it will be better. Examining will be easier.
Thank you for this comment Figure 10 was changed.
We would like to thank you once again for your suggestions for improving our manuscript.
Yours faithfully, Miklaszewski

Round 2
Reviewer 1 Report
The comments attached in file.

Author Response
Dear Reviewer
We would like to thank you for your comments and kind decision for us to resubmit our manuscript: Properties and microstructure evaluation in NiAl-xWC (x=0-90wt.%) intermetallic-based composites prepared by mechanical alloying to Materials. We are pleased with your comments and objective feedback considering the draft of our article. The changes were marked in our article in red colour.
All of the suggestions were included during the correction of our article and the comments were appropriately responded.
We would like to thank you once again for your suggestions for improving our manuscript.
Yours faithfully,
Miklaszewski
Reviewer 2 Report
The authors have replied to all comments and suggestions, and the necessary revisions have been made to the text. The manuscript may be published in the journal.
Author Response

(The authors gave the same response as above.)

Reviewer 3 Report
The authors have well addressed the comments. There is no additional comment.
Author Response

(The authors gave the same response as above.)
